# Bacterial Communities Associated with the Roots of *Typha* spp. and Its Relationship in Phytoremediation Processes

**DOI:** 10.3390/microorganisms11061587

**Published:** 2023-06-15

**Authors:** Joana Guadalupe Martínez-Martínez, Stephanie Rosales-Loredo, Alejandro Hernández-Morales, Jackeline Lizzeta Arvizu-Gómez, Candy Carranza-Álvarez, José Roberto Macías-Pérez, Gisela Adelina Rolón-Cárdenas, Juan Ramiro Pacheco-Aguilar

**Affiliations:** 1Facultad de Estudios Profesionales Zona Huasteca, Universidad Autónoma de San Luis Potosí, San Luis Potosí 79060, Mexico; 2Secretaría de Investigación y Posgrado, Centro Nayarita de Innovación y Transferencia de Tecnología (CENITT), Universidad Autónoma de Nayarit, Tepic 63173, Mexico; 3Facultad de Química, Universidad Autónoma de Querétaro, Santiago de Querétaro 76010, Mexico

**Keywords:** phytoremediation, *Typha* spp., bacterial diversity, *Proteobacteria*, heavy metal tolerance

## Abstract

Heavy metal pollution is a severe concern worldwide, owing to its harmful effects on ecosystems. Phytoremediation has been applied to remove heavy metals from water, soils, and sediments by using plants and associated microorganisms to restore contaminated sites. The *Typha* genus is one of the most important genera used in phytoremediation strategies because of its rapid growth rate, high biomass production, and the accumulation of heavy metals in its roots. Plant growth-promoting rhizobacteria have attracted much attention because they exert biochemical activities that improve plant growth, tolerance, and the accumulation of heavy metals in plant tissues. Because of their beneficial effects on plants, some studies have identified bacterial communities associated with the roots of *Typha* species growing in the presence of heavy metals. This review describes in detail the phytoremediation process and highlights the application of *Typha* species. Then, it describes bacterial communities associated with roots of *Typha* growing in natural ecosystems and wetlands contaminated with heavy metals. Data indicated that bacteria from the phylum *Proteobacteria* are the primary colonizers of the rhizosphere and root-endosphere of *Typha* species growing in contaminated and non-contaminated environments. *Proteobacteria* include bacteria that can grow in different environments due to their ability to use various carbon sources. Some bacterial species exert biochemical activities that contribute to plant growth and tolerance to heavy metals and enhance phytoremediation.

## 1. Introduction

Heavy metals (HMs) are chemicals used in many industrial processes, so their environmental levels have considerably increased, causing damage to living organisms. HM contamination is one of the most critical concerns due to its high persistence in the environment and its harmful effects on human health, plants, animals, and biodiversity [1]. So, physicochemical processes have been developed to remove HMs from polluted sites. However, they are expensive, inefficient, and non-ecofriendly. Therefore, phytoremediation alternatives have been designed to decrease the HM impact on the environment.

Phytoremediation is a sustainable strategy that uses plants to remove, reduce, transform, volatilize, concentrate, or stabilize HMs in soil, water, sludge, and sediments [2,3]. *Typha* is one of the most important genera used in phytoremediation due to its worldwide distribution in natural aquatic and semiaquatic ecosystems and its ability to persist in polluted environments [4]. *T. angustifolia*, *T. domingensis*, and *T. latifolia* are the most common plant species that remove HMs from water, soil, and sediments in natural and artificial wetlands [5,6,7]. They accumulate Zn, Ni, Pb, Cd, Cu, Al, Fe, and Mn mainly in their roots and, to a lesser extent, in aerial tissues [8,9,10]. Furthermore, *Typha* sp. removes organic pollutants such as synthetic pesticides, herbicides, and drugs, among other things [11]. Hence, as a versatile species with the potential for organic compound phytoremediation, it is the subject of study in this review.

It has been shown that the roots of plants that grow in HM-polluted environments are colonized by plant growth-promoting rhizobacteria (PGPR) that promote plant growth by reducing HM stress and improving phytoremediation [12]. Therefore, bacterial diversity associated with *Typha* spp. roots has attracted much attention because microorganisms contribute to plant adaptation and environmental conditions. Screening has been performed using culture-dependent techniques and 16S rRNA sequencing to identify bacteria associated with roots and shoots of *Typha* spp. exposed and non-exposed to HMs. Furthermore, plant–bacteria interaction assays have been established to determine the role of selected bacterial isolates when inoculated in plants exposed to HMs.

Therefore, this review shows the role of *Typha* sp. in phytoremediation processes, the microbial diversity associated with the roots of plants exposed to HM, and how bacterial communities could contribute to the plant adaptation to contaminants and their participation in phytoremediation.

## 2. Phytoremediation

Phytoremediation is derived from the Greek prefix *phyto* and the Latin *remedium*, meaning plants are used to correct or eliminate a xenobiotic compound [13]. The term phytoremediation was initially used to describe HM extraction from soils. However, it has been extended to plants used for organic compound removal, so concepts are shared to describe both HM and organic compound remediation [14]. Phytoremediation is an in-situ technique for the remediation of contaminated soils and water bodies, in which the plants remove, degrade, immobilize, neutralize, or contain pollutants. Plants and their associated rhizospheric microorganisms absorb, sequester, degrade, or metabolize contaminants through biological and physicochemical processes [15,16]. Phytoremediation is successfully used to clean up toxic metals (Ag, As, Cd, Cu, Co, Cr, Hg, Mo, Ni, Pb, and Zn), radionuclides (^90^Sr, ^137^Cs, ^239^Pu, ^234^U, and ^238^U), explosives, pesticides, and oils [17,18]. However, its use is limited by the climatic and geological site conditions where it is applied and by factors such as temperature, altitude, soil type, and accessibility for agricultural equipment [19].

### 2.1. Classification of Plants Used in Phytoremediation

HM-tolerant plants can grow in the presence of metals; however, they behave differently according to species and genotype [20]. Plants have been grouped into excluders, indicators, and accumulators according to the HM concentration they can store in their tissues, regarding the amount found in the soil (Figure 1) [20,21].

Excluder plants can tolerate the presence of high HM concentrations in the soil because they restrict the entry and translocation of HM to aerial tissues, which allows them to maintain low HM concentrations in aerial tissues regardless of the concentration in the soil [21,22,23]. Excluder plants include grass family members such as Sudan grass, barley grass, and fescue [20].

The indicator species are sensitive to HMs and can accumulate proportional HM concentrations to those found in the soil [21,22]. This group includes grain and cereal crops such as corn, soybeans, wheat, and oats [20].

Accumulator plants actively absorb metals from the soil and accumulate them in their aerial biomass in non-toxic forms. These plants can contain higher concentrations of HMs in their tissues than the HM concentration found in the soil [20,22]. Some accumulator plants can grow in soils highly contaminated with HMs and accumulate high concentrations in their aerial tissues without showing signs of toxicity. These plants have been called hyperaccumulators [24,25]. To be designated as a hyperaccumulator, a plant must accumulate > 10,000 mg/kg of Mn or Zn, >1000 mg/kg of Co, Cu, Pb, Ni, As, or Se, and >100 mg/kg of Cd [22,26].

The function of HM hyperaccumulation in plant tissue has been shown to serve as a defense against viruses, pathogenic fungi, and herbivores [21,27]. However, this characteristic has been exploited to remediate HM-contaminated soils through phytoremediation [25]. More than 450 hyperaccumulating species distributed among 45 families have been identified, which can accumulate even more than two types of HM [22,28,29].

### 2.2. HM Phytoremediation Mechanisms

The elimination or transformation of contaminants from soil, water, or sediments by phytoremediation is carried out using at least one of the following processes: phytoextraction, phytovolatilization, phytostabilization, rhizofiltration, phytodegradation, or phytostimulation (Figure 2). Each phytoremediation process uses a different contaminant treatment mechanism (Table 1).

Phytoextraction uses the ability of hyperaccumulator plants to absorb contaminants from the soil and transport, accumulate, and concentrate them in their aboveground biomass [30,31,32]. It removes contaminants from the soil, without affecting its structure and fertility, and treats wastewater [33,34]. This method extracts HMs, organic contaminants, and radioisotopes [33]. Hyperaccumulator plants can accumulate high levels of HM at levels 100-fold greater than non-hyperaccumulating without showing phytotoxicity symptoms. Some species from the *Brassicaceae*, *Fabaceae*, *Euphorbiaceae*, *Asteraceae*, *Lamiaceae*, and *Scrophulariaceae* families have been characterized as hyperaccumulators, and they are suitable for soil remediation [34].

In phytovolatilization, the plants take up soil contaminants, transform them into volatile forms, and eliminate them into the atmosphere through their leaves in less toxic forms [31,32]. It treats some organic compounds and toxic elements such as Hg, As, and Se in soil, sediment, or water [35]. These elements’ transformation occurs mainly in the root, and their release into the environment occurs during transpiration [36]. *Brassica juncea* from *Brassicaceae* family is a good Se volatilizer. Unfortunately, phytovolatilization only transfers pollutants from the soil to the atmosphere, contaminating the air with toxic volatile compounds [34].

Phytostabilization refers to using plants to immobilize contaminants in the soil to reduce their mobility and bioavailability for other plants or microorganisms [30,37]. Phytostabilization reduces leaching, runoff, and the distribution of pollutants to other areas by soil erosion [38,39]. It is accomplished through soil stabilization by roots or plant root exudates and the immobilization of contaminants through chemical and physical mechanisms such as precipitation, formation of insoluble complexes, reduction of valence, or adsorption [33,34,40].

Rhizofiltration is a technique used to remove contaminants from aquatic environments such as groundwater, surface water, and wastewater [41,42]. In this technique, the roots of plants are used to absorb, concentrate, and precipitate toxic metals and organic chemicals. The plants used can be both terrestrial and aquatic and have a high growth rate, in which the pollutants are absorbed and concentrated in their roots and stem [43]. Root exudates and changes in pH in the rhizosphere can also cause the precipitation of metals on root surfaces [43,44]. The most common aquatic plants used in rhizofiltration are hyacinth (*Eichhornia Crassipes*), azolla (*Azolla filiculoides*), duckweed (*Lemna minor*), cattail (*Typha* sp.), and poplar (*Populus* sp.) because of their fast growth, high biomass, and high tolerance and accumulation of HMs. Likewise, terrestrial plants such as Indian mustard (*Brassica juncea*) and common sunflower (*Helianthus annuus*) have good capacities for HM rhizofiltration [34].

Phytodegradation is used in the degradation of organic pollutants or the transformation of these to other less-toxic forms through the enzymatic activity of dehalogenases, oxygenases, and reductases of plants or rhizospheric microorganisms [43,45]. Through this mechanism, aromatic hydrocarbons, petroleum hydrocarbons, pesticides such as herbicides, insecticides, and fungicides, chlorinated compounds, explosives such as trichloroethylene, and detergents can be degraded [32,43,45,46].

Phytostimulation is a phytoremediation mechanism in which substances exuded from plant roots stimulate the growth of microorganisms capable of degrading organic contaminants [33,47].

Overall, the best phytoremediation mechanism is exerted by hyperaccumulator plants for permanent HM removal. However, most hyperaccumulators are short-lived, with low biomass production and slow growth rates, limiting their phytoextraction efficiency. Thus, high biomass-producing non-hyperaccumulators can be used as alternatives. Although non-hyperaccumulator plants accumulate lower HM concentrations in their aboveground tissues, their high biomass production reaches HM accumulation levels similar to those of hyperaccumulators [34].

Therefore, the rhizofiltration technique is applied in constructed wetlands to remove HMs and organic pollutants from groundwater, surface water, and wastewater. Some species of *Scirpus*, *Phagmites*, *Eichhornia*, *Azolla*, *Lemna*, and *Typha* genera have been studied and have demonstrated their abilities to reduce the concentration of contaminants in water or sediment. Among these, *Typha* stands out due to its capacity to adapt to nutrient deprivation, flood, salinity, and drought. Moreover, high biomass production, fast growth, and high HM tolerance and removal make the *Typha* species an excellent option for removing HMs from wetland wastewater.

## 3. *Typha* Genus

The *Typha* genus, commonly known as “Cattail”, is distributed worldwide, except in Greenland and Antarctica. Plants belonging to this genus are erect, rhizomatous, perennial herbs that flourish on a slender stem up to 3 m high [48]. The flowers are small and clustered in spikes, while the creeping lateral rhizomes, or underground stems, reach up to 70 cm in length and 3 cm in diameter. The leaves are lanceolate, distichous, flat, long, and grayish-green [49,50]. The *Typha* genus includes nine species: *T. minima*, *T. elephantine*, *T. angustifolia*, *T. domingensis*, *T. capensis*. *T. latifolia*, *T. shuttleworthii*, *T. orientalis*, and *T. laxmannii* [48].

*Typha* species grow in bogs, swamps, wetlands, roadside ditches, lakeshores, pond shores, irrigation canals, and ponds agricultural irrigation ditches [50,51,52]. Although *Typha* species are ecologically important, they can also be considered an invasive native species in aquatic ecosystems because of their high growth rate and ability to adapt to saline environments, nutrient deficiencies, floods, and drought [48,49,53].

Moreover, *Typha* species have an essential role in wetland biogeochemistry since they are the main producers and drivers of the organic matter cycle, directly affecting wetland biodiversity because it is the habitat of several animal species and microorganisms [54]. Wetlands represent one of the most significant biological carbon (C) pools [55] and have received much attention for their potential for the sequestration and long-term storage of substantial amounts of atmospheric CO_2_ and climate change mitigation [56,57]. The *Typha* sp. has played an essential role in this ecosystem service. Eid and Shaltout [55] concluded that Lake Burullus in Egypt, associated with *T. domingensis* plants, has the potential to sequester C. Additionally, in *T. angustifolia*, it has been reported that the amount of biomass, and therefore the efficiency of carbon sequestration, is higher than in other macrophytes [57,58].

Although wetlands provide an optimal environment for CO_2_ sequestration, they are sources of greenhouse gas (GHG) emissions, mainly methane (CH_4_) [56], which is documented in restored wetlands with *Typha* plants in California, where significant CH_4_ emissions were recorded while sequestering C [59]. High water levels have been associated with higher CH_4_ emissions in wetlands planted with *T. latifolia* due to aerenchyma, which allows it to grow in flooded environments and consequently increases CH_4_ emissions [60,61]. Therefore, it is crucial to understand the water level dynamics of wetlands with *T. latifolia* and other *Typhaceas* to regulate CO_2_ and CH_4_ fluxes and achieve ecosystem balance. On the other hand, paludiculture (perennial crops growing in humid or re-humidified agricultural peatlands) has been considered a solution to GHG emissions. The paludiculture with *T. latifolia* has effectively improved water quality and reduced GHG (such as CH_4_) emissions in rewet peatlands [60,62].

The sequestration of C in *Typha* spp. is determined by the proportion of leaf, stem and root biomass, structure and morphology, water levels, restoration design, wetland age, and disturbance events [57,58,61,63]. Therefore, these factors must be considered to understand the balance between CO_2_ and CH_4_ exchanges in wetlands with *Typha* sp. Furthermore, there is a need to protect these wetlands, which are essential for carbon sequestration and other ecosystem services to reduce climate change.

### Typha *spp.* Applications in Phytoremediation

Plants belonging to the *Typha* genus can quickly colonize wetlands through their rhizomatous propagation, rapid growth rate, and high capacity to adapt to various environmental conditions [49]. They are used in artificial and natural wetlands for wastewater treatment, and thus are essential in maintaining ecosystem health [64].

*Typha* sp. can perform water depuration processes removing contaminants, including HMs, nutrients (nitrates, phosphates, etc.), solvents, explosives, crude oil, organic pollutants, and pesticides [11,65,66]. So, it removes contaminants from aquatic ecosystems, allowing wildlife development and protecting coasts from erosion and marine environments [49].

*Typha* species have been successfully used in phytoremediation strategies due to their growing ability in HM-polluted environments [1,53]. Their remarkable efficiency in removing metallic ions lie in their airy internal structure, comprised of tissues with open spaces, allowing better contaminant absorption [67]. *T. latifolia*, *domingensis*, and *T. angustifolia* are the most common species used in phytoremediation due to their rapid growth, large biomass amount, and HM tolerance abilities [7]. They are employed in natural and constructed wetlands to eliminate HMs from polluted water, accumulating mainly in the roots and scarcely in their aerial tissues (Table 2). These abilities are because most *Typha* species possess a bioconcentration factor (BCF) > 1, indicating the remarkable capacity to remove HMs from water, sediments, and HM solutions and accumulate them in their roots. On the other hand, *Typha* species have a translocation factor (TF) < 1, meaning a low capacity to translocate HMs to aerial tissues. Plants with a BCF greater than one and a TF less than one can be used for rhizoremediation due to their phytostabilization potential [7]. So, *Typha* species have been used in tertiary wastewater treatment due to their remarkable ability to accumulate toxic elements in their root system by rhizofiltration [68].

*Typha* sp. has been used to treat wastewater containing high concentrations of organic matter, nitrogen, sulfur compounds, and chromium from tanneries [69]. *T. domingensis*, *T. latifolia*, and *T. angustifolia* have been used to remove Al, As, Cd, Cr, Cu, Hg, Mn, Ni, Pb, and Zn from a natural wetland containing municipal wastewater and HM contamination [7]. Likewise, Carranza-Álvarez et al. [70] demonstrated that *T. latifolia* removed Pb, Cd, Cr, Mn, and Fe from Tanque Tenorio, an artificial lagoon highly polluted by municipal and industrial wastewater. Klink et al. [71] showed that *T. latifolia* removed Pb, Cu, Co, and Zn from small ponds and accumulated them in their roots and rhizomes. The HM tolerance, high element uptake ability, and the excellent biomass production of *Typha* species make them the best species for the phytoremediation of HM-contaminated environments.

**Table 2 microorganisms-11-01587-t002:** Metal accumulation in various plant parts of *Typha* species.

Species	Site/System	Heavy Metal	Metal Concentration (mg/Kg)			References
			Shoots/Leaves/Stems	Roots	Rhizome	
*T. latifolia*	Constructed wetland	Zn	59.29	177.28	NR	[5]
		Cu	14.73	33.29	NR	
	Natural wetland	Al	38.3–48.5	1740–1780	845–1055	[7]
		As	0.08–0.12	1.87–2.21	1.21–1.65	
		Cd	0.06–0.08	0.39–0.46	0.16–0.22	
		Cr	0.95–1.01	5.54–6.75	3.24–3.85	
		Cu	4.66–5.87	12.8–13.1	9.87–11.8	
		Hg	0.49–0.63	2.88–3.35	1.55–1.83	
		Mn	29.7–41	132–155	70.1–103	
		Ni	8.42–10.3	35.6–41.2	28.5–30.2	
		Pb	0.44–0.52	13.5–15.2	4.32–6.65	
	Stream	Zn	215	340	NR	[9]
		Ni	40	55	NR	
		Cu	30	50	NR	
		Pb	8	13	NR	
		Co	10	24	NR	
		Mn	990	860	NR	
		Cd	0.21	0.44	NR	
		Cr	21	44	NR	
	Artificial lagoon	Zn	28.7–41	110–115	96.5–103	[70]
		Cd	0.1–1.85	0.1–25	NR	
		Cr	1–32	1–60	NR	
		Mn	63–1162.5	125–2375	NR	
		Fe	130–375	325–500	NR	
	Pond	Fe	178	8431	1875	[71]
		Mn	477	1943	292	
		Zn	28	373	65.6	
		Cu	3	8.62	3.97	
		Cd	0.01	7.28	2.72	
		Pb	3.0	12.1	6.33	
		Ni	3.7	27.8	8.92	
		Co	0.25	2.57	0.96	
		Cr	6	35.7	11.7	
	Constructed wetland	Cd	276–622	932–2339	NR	[72]
		Pb	272–927	1365–4867	NR	
	Natural wetland	Cu	16.00	13–265	37	[73]
		Ni	54	388	80	
		Zn	8–67	24–572	23,894	
		Fe	114–504	777–57,138	105–17,162	
		Mn	64–1734	16–901	16–552	
		Mg	564–2550	882–5542	745–2872	
		Ca	2687–16,993	1781–11,574	1209–6726	
	Constructed wetland	Fe	25–91	650–1250	NR	[74]
		Cu	15–49.98	10–31.45	NR	
		Pb	2.5–3.95	45,049	NR	
		Hg	2.5	45,082	NR	
		Zn	11,871	15–35	NR	
	Constructed wetland	Cu	13.52	32.92	NR	[75]
		Cd	11.84	14.68	NR	
		Mn	50.26	32.14	NR	
		Cr	11.46	10.72	NR	
		Co	8.28	11.1	NR	
		Zn	123.7	102.9	NR	
		Pb	19.38	24.38	NR	
		Ni	7.4	11.82	NR	
	River	Cd	0.89	1.1	NR	[76]
		Ni	1.955	26.9	NR	
		Zn	9.66	98.1	NR	
		Cu	4.885	30.2	NR	
	Constructed wetland	Pb	NR	65.6	NR	[77]
		Cr	NR	22.1	NR	
		Mn	NR	219	NR	
	Constructed wetland	As	0.001–0.02	0.008–0.03	NR	[78]
		Cd	17–118	185–319	NR	
		Cr	2.84	37–99	NR	
	Lakes	Fe	58.55	1252	125	[79]
		Pb	4.365	1.07	7.79	
		Mn	127.85	536	115	
		Cd	0.075	2.76	0.14	
		Cu	3.185	11.6	4.19	
		Ni	0.72	9.42	3.14	
		Zn	18.9	77.6	58.4	
	Constructed wetland	Cd	26.1–131	50.9–279	NR	[80]
*T. domingensis*	Constructed wetland	Fe	63.23	40.6	NR	[6]
		Mn	8.59	28.88	NR	
		Ni	4.8	24.3	NR	
		Pb	0.51	7	NR	
		Cr	8.17	17.6	NR	
	Natural wetland	Al	38–50.9	1756–1890	850–920	[7]
		As	0.08–0.10	2.78–3.21	1.29–1.34	
		Cd	0.05–0.08	0.44–0.61	0.15–0.18	
		Cr	1.05–1.24	3.67–5.88	3.01–4.57	
		Cu	3.50–4.67	15.2–18.5	10.4–12.7	
		Hg	0.85–0.97	3.21–3.67	2.02–2.56	
		Mn	32.1–51.2	138–151	74.2–83.8	
		Ni	10.8–10.9	36.6–53.3	29.6–38.7	
		Pb	0.65–0.71	10.9–13.7	4.21–4.33	
		Zn	35.4–38.8	118–122	97.3–103	
	Natural wetland	Ba	75.6	51.57	NR	[81]
	Natural wetland	Cd	1.25–21.3	188.62–234.10	NR	[82]
	Constructed wetland	Cr	10–90	50–750	10–300	[83]
		Ni	10–60	100–800	10–250	
		Zn	15–60	50–150	10–50	
	River	Hg	0.0506–0.5604	0.9785–5.474	0.4238–1.802	[84]
	Plastic reactor	Cr	2200–4000	3500–7000	200–1500	[85]
		Ni	1400	500–1000	200–500	
		Zn	2350–4750	300–3000	100–500	
	Constructed wetland	Ba	41.85–1398	303.15–3795.27	NR	[86]
	Pond	Al	187–282	220.82–350.55	NR	[87]
		Fe	102–173	307.5–582.44	NR	
		Zn	11.49–57	28.06–149.60	NR	
		Pb	1.7–9.0	1.26–20.46	NR	
	Constructed wetland	Hg	0.1785–273.3515	NR	NR	[88]
	Constructed wetland	Cr	NR	82	NR	[89]
		Ni	12	66	NR	
		Zn	28	178	NR	
*T. angustifolia*	Natural wetland	Al	36.1–44.6	1568–1865	821–962	[7]
		As	0.05–0.06	1.95–2.86	1.06–1.42	
		Cd	0.04	0.38–0.51	0.10–0.20	
		Cr	0.75–0.91	4.26–5.15	1.89–2.48	
		Hg	0.35–0.55	1.98–2.75	1.01–1.96	
		Mn	31.6- 36	95.8–126	77.6–103	
		Ni	8.96–12.3	28.8–35.7	20.2–21.6	
		Pb	0.52–0.75	8.90–10.2	3.25–5.23	
	Constructed wetland	Zn	33.9	37	NR	[90]
		Cd	7.3	7.2	NR	
		Pb	0.8	2.8	NR	
	Constructed wetland	Cd	20.3–42.3	241–378.3	NR	[91]
		Pb	354.9–1875.9	20,173.6–22,462	NR	
	Constructed wetland	Cd	0.225	0.82	NR	[92]
		Cr	8.345	59.13	NR	
		Cu	8.55	35.14	NR	
		Fe	701.375	3327	NR	
		Ni	4.025	21.1	NR	
		Pb	10.465	50.82	NR	
		Zn	100.075	150	NR	
	Natural wetland	Cd	0.03–0.65	0.1–0.8	NR	[93]
		Pb	0.3–4.5	1–6	NR	
		Cr	0.75–7.75	1.5–7.5	NR	
		Ni	1.75–16.25	2.5–15	NR	
		Zn	20–70	10–100	NR	
		Cu	0.75–25.5	2.5–17.5	NR	
	Constructed wetland	Pb	57.8–167.3	1265.2–8937.4	68.7–158.9	[94]
	Hydroponics	Cr	234.02–1157.28	287.16–4399.79	NR	[95]
*T. capensis*	Natural wetland	Cr	69–3560.5	222–16,047	70–786	[96]
		Fe	3176.5–8511.5	9413–13,833	2303–8970	
		Zn	21–59	56–162	24–30	
		Cu	13–31	35–224	10–56	
		Co	11–29	58–124	5–10	
		Cd	23.5–26.5	16–22	18–21	
		Ni	29–44	196–891	17–88	
		Pb	7.5–54.5	27–63	6–16	

NR: Data not reported.

## 4. Bacteria Associated with the Rhizosphere of *Typha* spp.

The *Typha* genus has a root zone rich in dissolved oxygen and organic carbon that provides favorable conditions for colonization by microorganisms [97]. Plants’ root exudates are essential in the selection and abundance of bacterial communities at the rhizoplane and endosphere roots [98]. Moreover, the composition and structure of bacterial communities are determined by environmental factors, including the type of wetland (natural or artificial), water quality, soil composition, pH, geographical location, root zone, plant species, plant phenological phase, stress, and disease events [99,100]. So, rhizospheric bacterial communities can vary in the diversity and abundance of species across the plants, even between plants belonging to the same species. In all ecosystems, bacterial communities are adapted to the plant rhizosphere conditions, where they establish a range of beneficial and deleterious interactions among themselves. Moreover, these bacterial populations contribute to development, growth, and plant adaptation to ecosystems (Figure 3).

### 4.1. Bacterial Communities Associated with Typha Roots in Natural Environments

A few studies have been explored bacterial communities associated with *Typha* roots growing in natural environments. Bacteria have been identified by traditional microbiology techniques or by 16S rRNA gene sequencing (Appendix A).

Jha and Kumar [101] isolated ten endophytic diazotrophic bacteria from *T. australis* roots collected from ditches of the agricultural farms at Banaras Hindu University (Figure 4). Bacteria were characterized according to their plant growth-promoting abilities, with *Klebsiella oxytoca* GR-3 being the most efficient because of its nitrogenase activity, IAA production, and phosphate solubilization. A plant–bacteria interaction assay showed that *K. oxytoca* GR-3 promotes the growth of rice seedlings by increasing the root and shoot length, fresh weight, and chlorophyll content [101]. Thus, root-endophytic *K. oxytoca* GR-3 was the first PGPR isolated from the *Typha* genus. Despite its plant growth-promoting potential, the effect of *K. oxytoca* GR-3 in its host, *T. australis*, remains unknown.

Likewise, Ashkan and Bleakley [102] isolated nine cultivable endophytic bacteria from the roots of *Typha* sp. collected at streams of Hot Springs, South Dakota. Then, 16S rRNA sequencing identified seven *Bacillus* sp. (*Firmicutes*) and two *Pseudomonas* sp. (*Proteobacteria*) (Figure 4). These studies were limited because the screening was focused on cultivable bacteria. Furthermore, the plant growth-promoting abilities of the isolates were not tested, so their role in *Typha* sp. is unknown.

On the other hand, PCR-guided analysis of the 16S rRNA gene was performed to elucidate the composition of the root-associated bacterial microbiota of *T. latifolia* growing in a wetland in the natural paradise “Le Mortine Oasis”, Campania, southern Italy [65]. The root-associated microbiota of *T. latifolia* was dominated by the phylum *Proteobacteria* (31 isolates, 39 %), followed by *Actinobacteria* (20 isolates, 25%), *Firmicutes* (12 isolates, 15%), *Planctomycetes* (8 isolates, 10%), *Acidobacteria* (5 isolates, 6%), and *Chloroflexi* (3 isolates, 4%) (Figure 4). Among them, *Proteobacteria* dominate rhizoplane, rhizosphere, and at least a 2 m radius of surrounding water of *T. latifolia* collected in the wetland. These results agree with previous data obtained for the plant microbiota, mainly dominated by *Proteobacteria* [103]. *Rhizobiales*, *Rhodobacterales*, and *Pseudomonadales* were the most abundant *Proteobacteria* associated with *T. latifolia* roots [65]. This is due to *Proteobacteria* being fast-growing r-strategists that are able to utilize a broad range of root-derived carbon substrates, showing adaptation to the diverse plant rhizospheres [103,104]. The most abundant phyla, *Proteobacteria*, *Actinobacteria*, and *Firmicutes*, include bacterial species of the families *Rhizobiaceae*, *Pseudomonadaceae*, *Streptomycetaceae*, and *Bacillaceae* that potentially promote plant growth [103]. These bacterial species could be involved in adapting *T. latifolia* to the environmental conditions in the “Le Mortine Oasis”. These results indicated that sequencing analysis generated a broader overview of bacteria associated with plant roots and is an excellent strategy for exploring bacterial diversity and abundance in the plant rhizosphere.

Cultivable bacteria from *T. latifolia* rhizoplane were also isolated, and we determined their biofilm formation capacity [65]. The best biofilm-forming bacteria were identified by 16S rRNA gene sequencing, revealing the presence of *Microbacterium chocolatum*, *Streptomyces* sp., *Streptomyces mirabilis*, *Rhodococcus* sp., *Wautersiella* sp., *Pseudomonas* sp., *Janthinobacterium lividum,* and family members of *Xanthomonadaceae*, *Enterobacteriaceae*, and *Bacillaceae,* which colonize the *T. latifolia* rhizoplane. However, the PGPR activities of bacterial isolates were not determined, as they could contribute to the adaptation and growth of *T. latifolia* in the “Le Mortine Oasis” [65]. Overall, combining 16S rRNA gene sequencing and isolation by culture-dependent techniques offers a broader view of microbial populations associated with *T. latifolia* roots (Appendix A). Even bacterial isolates or consortia could be tested in interaction assays with their host to identify the most efficient isolates that promote the growth of *T. latifolia*.

### 4.2. Bacterial Communities Associated with the Roots of Typha Exposed to HMs

*Typha* species have been used in phytoremediation because of their abilities to remove HMs from the surrounding environment [1,53,66]. It has been demonstrated that HM removal is enhanced by microorganisms associated with the plant roots [105]. So, identifying bacterial populations associated with the roots of *Typha* species has gained importance (Table 3; Appendix A). However, available information has been obtained from *Typha* species collected in different environmental conditions and using either culture-dependent or gene sequencing strategies (Figure 5). So, the information on specific bacterial communities of each *Typha* species is scarce for defining the *Typha* microbiome.

The first study with a *Typha* species was conducted by Pacheco-Aguilar et al. [69], who isolated bacteria that colonize the roots of *Typha* sp. growing in an artificial wetland to treat the wastewater of a tannery contaminated with high concentrations of organic matter, chromium, nitrogen, and sulfur compounds. Eight bacterial strains of the phylum *Proteobacteria* were isolated, including three *Pseudomonas* sp., two *Acinetobacter* sp., two *Alcaligenes* sp., and one *Ochrobactrum* sp. (Table 3; Appendix A). The characteristics of the wastewater agree with the predominance of bacteria belonging to the phylum *Proteobacteria,* which could be involved in degrading organic matter.

It is known that the versatile *Pseudomonas* genus can tolerate HMs because of its ability to adapt to various environmental conditions [108,109]. *Ochrobactrum* strains have been shown to alleviate Cd toxicity in spinach (*Spinacia oleracea* L.) [110] and rice (*Oryza sativa*) [111] and adsorb Cd and Cr in their cell wall due to exopolysaccharides content [112,113,114]. Similarly, some *Acinetobacter* strains have been applied to improve the removal efficiency of heavy metals such as Cr [115,116,117], while *Alcaligenes* has been reported to be tolerant to Cd, Cu, Ni, Zn, and Cr [118,119].

It is probable that *Pseudomonas* sp., *Acinetobacter* sp., *Alcaligenes* sp., and *Ochrobactrum* sp. play an essential role in adapting *Typha* sp. to wetland conditions and improving the Cr removal by the plant.

Shehzadi et al. [106] isolated cultivable endophytic bacteria from the shoots and roots of *T. domingensis* grown in wetlands to treat textile effluents contaminated with organic matter, phosphates, sulphates, nitrates, and Ni, Fe, Cr, and Cd. These endophytic bacteria belong to the phyla *Proteobacteria*, *Firmicutes*, *Actinobacteria*, and *Bacteroidetes* (Table 3; Appendix A). Among them, *Bacillus* sp. TYSI17, *Microbacterium arborescens* TYSI04, *Microbacterium* sp. TYSI08, *Rhizobium* sp. TYRI06, *Pantoea* sp. TYRI15, and *Pseudomonas fluorescens* TYSI35 exerted PGPR activities and exhibited the best textile effluent degrading activity. These results suggest that bacterial strains could be involved in the growth and adaptation of *T. domingensis* to the wetland conditions. The most efficient bacterial strains could be used to remediate industrial effluents.

Saha et al. [64] isolated ten cultivable endophytic bacteria from the roots of *T. angustifolia* grown at a uranium mine tailing contaminated with iron. The 16S rRNA gene sequencing of the isolates indicated that the endophytic bacteria belonged to the phyla *Firmicutes*, *Proteobacteria*, and *Actinobacteria* (Table 3; Appendix A). All bacterial isolates exerted biochemical activities, including auxin synthesis, siderophore production, and nitrogen fix. Plant–bacteria interaction assays showed that a consortium including ten isolates promotes the growth of *T. angustifolia* and rice seedlings, indicating that isolates are PGPR, which could participate in the adaptation of *T. angustifolia* to the contaminated site. Thus, the consortium has the biotechnological potential to be applied for plant-bacteria remediation purposes.

Additionally, Zhou et al. [107] investigated the composition and abundance of ammonium-oxidizing bacteria in the rhizosphere of *T. orientalis* exposed to Cu, Cr, Pb, Cd, and Zn on the shores of Jinshan Lake Park in China, and 16S rRNA sequencing revealed that the bacterial communities belonged to the phylum *Planctomycetes* (Table 3; Appendix A). The bacterial composition of the rhizosphere was affected by the presence of Cu, Pb, and Zn, the availability of nitrogen, and even the stage of growth and the year’s season. The phylum *Planctomycetes* has been associated with soil microbial communities in response to stress from heavy metals, such as Hg, Cd, and Cr [120,121]. This phylum can detoxify heavy metals by secreting extracellular substances, such as polysaccharides and proteins, reducing the toxic effects of metals [122,123]. The increase in the *Planctomycetes* population has also been related to an acidic pH, and this phylum’s diversity varies according to environmental conditions [120,121].

Rolón-Cárdenas et al. [53] isolated four endophytic strains of *Pseudomonas rhodesiae* from the roots of *T. latifolia* exposed to Cd in a contaminated site. Bacterial strains showed high tolerance to Cd and exerted biochemical activities such as PGPR (Table 3; Appendix A). Plant–bacteria interaction assays showed that *P. rhodesiae* strains promote the growth of *Arabidopsis thaliana* seedlings exposed and non-exposed to Cd. Likewise, *P. rhodesiae* decreases oxidative stress in *T. latifolia* seedlings exposed to Cd and improves Cd translocation to the shoot in an axenic hydroponic system [124]. Additionally, Rubio-Santiago et al. [125] isolated endophytic bacteria *P. azotoformans*, *P. fluorescens*, *P. gessardii*, and *P. veronii* from the roots of *T. latifolia* growing at a Pb- and Cd-contaminated site. Bacterial strains exerted biochemical activities such as PGPR and promoted the growth of *T. latifolia* seedlings exposed and non-exposed to either Pb or Cd. Thus, these bacterial strains could be used in plant–bacteria interactions for phytoremediation with *T. latifolia*.

The studies of microorganisms associated with the roots of the *Typha* genus were focused on isolating PGPR tolerant to HMs, because they promote plant growth and improve phytoextraction capacity in contaminated environments. However, the mechanisms by which endophytic bacteria enhance phytoremediation in *Typha* remain unclear. Furthermore, no *Typha*-bacteria model is available to determine the global mechanisms involved in phytoremediation.

## 5. Plant Growth-Promoting Rhizobacteria Associated with HM-Tolerant Plants

The rhizosphere is a narrow zone of soil that extends at least 2 mm from the root surface and contains exudates, including organic acids, sugars, amino acids, small peptides, and secondary metabolites released by the root plants. It has been shown that the rhizosphere is colonized by 10^8–^10^12^ bacteria per gram of soil, approximately a thousand-fold higher than in bulk soil [100,126]. Rhizospheric bacteria are essential in organic matter transformations and biogeochemical cycles [103]. So, the bacteria’s role in the rhizosphere can be beneficial, detrimental, or neutral for plants according to the soil conditions [127].

Plant growth-promoting rhizobacteria (PGPR) are found in the plant rhizosphere, where they promote plant growth through different mechanisms, including indole acetic acid (IAA), siderophores production, phosphate solubilization, and reducing plant stress by the action of 1-aminocyclopropane-1-carboxylic acid (ACC)-deaminase, which interferes in ethylene biosynthesis [128]. Overall, PGPR mechanisms are involved in plant growth in natural environments and plants exposed to stress by salinity, drought, organic compounds, and HMs [129].

The study of plant–PGPR interaction for HM removal has recently increased (Table 4). PGPR can colonize the roots of plants exposed to HMs, where they modify plant physiology by modulating hormonal status, restoring photosynthetic pigment synthesis, and increasing the synthesis of antioxidants such as phenolic compounds, glutathione, soluble sugars, and proline, which reduce oxidative stress in plants [105,124]. Furthermore, PGPR improve the processes for metabolism, extraction, and accumulation of HMs in plant tissues through the mechanisms described below [102,130].

### 5.1. Indole Acetic Acid’s Role in Plant Tolerance to HMs

PGPR can synthesize IAA and related compounds from L-Trp by five biosynthetic pathways [131]. IAA synthesized by bacteria is a strategy to colonize plant roots since it increases root growth and elongation, providing a larger surface to colonize and absorb nutrients [132]. PGPR can promote plant root growth by regulating auxin biosynthetic pathways in its host plant. For instance, Wu et al. [105] demonstrated that Cd-tolerant endophytic *Pseudomonas fluorescens* colonizes the root system of its host, upregulating genes involved in IAA synthesis that promote adventitious roots emergence in *Sedum alfredii* exposed to Cd. Furthermore, *P. fluorescens* increases the Cd-removal capacity of *S. alfredii*, suggesting that interaction improves the phytoextraction process.

IAA-producing bacteria have been shown to restore photosynthetic pigment production in plants exposed to HM stress. IAA-producing bacteria *Sphingomonas* sp. [133], *P. fluorescens* [134], and *Buttiauxella* sp. [135] increase the chlorophyll content in *S. alfredii*, while *Serratia* sp. increases it in *L. usitatissimum* [136]. Furthermore, IAA-producing bacteria decrease plant oxidative stress by inducing antioxidant compound synthesis. More details are available in Rolón-Cárdenas et al. [124], who reported the effect of auxin and auxin-producing bacteria in plant tolerance and accumulation of Cd.

### 5.2. Siderophore’s Role in Plant Tolerance to HMs

Siderophores are low-molecular-weight molecules that chelate iron with a very high and specific affinity. Gram-positive and Gram-negative bacteria secrete them to scavenge iron from their extracellular environment. Siderophore–iron complexes are transported into the cell through specific receptors in the bacterial membrane [137]. Diazotroph microorganisms, plant pathogens, and PGPR produce these compounds. The presence of HMs and nutrient deficiency stimulate their production. It has been shown that PGPR produce siderophores mainly in HM stress conditions. Thus, they can increase metal solubility by releasing siderophores in the rhizosphere [138].

Siderophore-producing PGPR have been isolated from *S. alfredii* and *Withania somnifera* growing in Cd-contaminated soils [139,140]. Sinha and Mukherjee [141] demonstrated that *Pseudomonas aeruginosa* KUCd1 increases pyoverdine excretion when exposed to high Cd concentrations. Likewise, high Cd, Al, Cu, and Ni levels induce siderophore production in *Streptomyces* sp. [142]. Increased bacterial siderophore production reduces free metal ions’ toxicity since siderophore–HM complexes prevent metal from entering the cell [143,144].

Although their primary function is to chelate ferric iron, siderophores can bind to metallic ions such as Cr, Al, Cu, Cd, Eu, and Pb, playing an essential role in HM removal from contaminated sites, thus becoming a beneficial ecological agent for their remediation. These compounds increase metals’ solubility, availability, and accumulation while decreasing toxicity. They also stimulate plant growth in contaminated sites and, in some cases, reduce the absorption of metals [145,146] (Table 4).

### 5.3. Phosphate Solubilization’s Role in Plant Tolerance to HMs

HMs in the soil interfere with iron and phosphorus absorption, so their deficiency causes slow plant growth and decreased biomass. The solubilization of inorganic phosphates and siderophores production by bacteria can compensate for this limitation. PGPR produce low-molecular-weight organic acids to solubilize phosphate from HM-polluted soils, thus contributing to phosphorus bioavailability for plant nutrition. Phosphate solubilization is carried out by the action of organic acids synthesized by bacteria such as oxalic, citric, butyric, malonic, lactic, succinic, malic, gluconic, acetic, glycolic, fumaric, adipic, and 2-ketogluconic acid [147] (Table 4). Phosphate-solubilizing PGPR have been approached in phytoremediation because organic acids synthesized by them chelate cations and thus improve phytoextraction by metal mobilization and accumulation in plant tissues [148]. Additionally, Teng et al. [149] demonstrated that in addition to organic acids synthesis, acid phosphatase activity in *Leclercia adecarboxylata* and *Pseudomonas putida* was involved in phosphate solubilization activity.

Furthermore, phosphate-solubilizing bacteria have been shown to improve phytostabilization by decreasing metal toxicity by transforming metal species into immobile forms. For instance, *L. adecarboxylata* can complex lead ions into hydroxyapatite (Pb_10_(PO_4_)_6_(OH)_2_) and pyromorphite (Pb_5_(PO_3_)_3_Cl), indicating its potential for Pb immobilization [149].

The phytoremediation of metals associated with phosphate-solubilizing PGPR has been shown to overcome drawbacks imposed by metal stress on plants [148]. Likewise, it improves the phytoextraction and phytostabilization of HMs; hence, this bacterial group could be used in phytoremediation strategies.

### 5.4. ACC Deaminase’s Role in Plant Tolerance to HMs

Ethylene is an important plant phytohormone involved in both growth and senescence, playing a pivotal role in accomplishing the plant life cycle. However, when plants are exposed to biotic and abiotic stress, it is produced in high amounts as a defense mechanism to inhibit plant growth in adverse environmental conditions [150,151,152].

The presence of HMs in soils causes the increase of ethylene, which negatively affects the growth of plants exposed to them [150,151]. However, PGPR with ACC-deaminase activity hydrolyze the precursor of ethylene biosynthesis, the 1-amino cyclopropane carboxylic (ACC), producing α-ketobutyrate and ammonia. Thus, this reduces ethylene levels and stress in plants, therefore restoring plant growth in HM-contaminated environments [153] (Table 4). Evidence demonstrates that PGPR possessing ACC-deaminase activity improves plants’ growth in the presence of HMs. For instance, Grichko et al. [154] demonstrated that the transgenic tomato plant *Lycopersicon esculentum,* expressing the bacterial gene ACC deaminase, tolerates Cd, Co, Cu, Mg, Ni, Pb, or Zn and accumulates greater amounts than non-transgenic plants. Moreover, it has been established that PGPR with ACC-deaminase activity improve the growth of plants in adverse environmental conditions, including heat, cold, drought, flooding, nutrient deficiency, phytopathogens, and pest attacks [155].

**Table 4 microorganisms-11-01587-t004:** Plant Growth-Promoting Rhizobacteria associated with HM-tolerant plants.

Heavy Metal	Bacterium	Plant	PGPR Activities	Bacterial Effects on Plants	References
Cd^2+^	*Serratia* sp. strain CP-13rif	*Linum usitatissimum* L.	Phosphate solubilization, IAA production, and ACC deaminase activity.	Bacterium enhances biomass accumulation and the roots and shoots growth. It increases photosynthetic pigments (Chl a, Chl b, and Chl total), proline, phenolic compounds, protein content, CAT activity and reduces H_2_O_2_ and MDA levels.	[136]
Cd^2+^	*Raoultella* sp. strain X13	*Brassica chinensis* L.	Phosphate solubilization, IAA, and siderophore production.	Bacteria enhance fresh and dry biomass accumulation and increase the content of soluble sugars.	[156]
Cd^2+^	*Cupriavidus necator* strain GX_5	*Brassica napus*	Siderophore secretion, ACC deaminase, IAA, and hydrogen cyanide (HCN) production.	Bacterium enhances dry biomass accumulation and root growth.	[157]
	*Sphingomonas* sp. strain GX_15		IAA production.		
	*Curtobacterium* sp. strain GX_31		ACC deaminase, IAA, and HCN production.		
Cd^2+^	*Kocuria rhizophila* strain 14asp	*Glycine max* L.	Phosphate solubilization, catalase activity, ACC-deaminase, IAA, and ammonia production.	Bacterium enhances the growth of the shoots.	[158]
Cd^2+^	*Serratia marcescens* strain S2I7	*Oryza sativa*	Phosphate solubilization, production of siderophore, IAA, and HCN.	Bacterium increases shoot growth and root length.	[159]
Cd^2+^	*Sphingomonas* sp. strain SaMR12	*Sedum alfredii*	Siderophore production, phosphate solubilization, IAA production.	Bacterial inoculation increases photosynthetic pigments (Chl). It decreases H_2_O_2_ and MDA levels in roots. In shoots, it downregulates the SaZIP2 gene, whereas it upregulates *SaZIP3*, *SaNramp6*, *SaHMA2*, and *SaHMA3* genes. In roots, the bacterium upregulates *SaZIP3* and *SaNramp1* genes and downregulates the *SaNramp3* gene.	[133,160,161]
Cd^2+^	*Pseudomonas fluorescens* strain Sasm05	*Sedum alfredii*	IAA production, siderophore production, and ACC deaminase activity.	Bacterium enhances biomass accumulation, promotes shoots, and root formation and increases photosynthetic pigments (Chl). In shoots, it upregulates *SaHMA2*, *SaHMA3*, *SaNramp1*, *SaNramp6*, *SaZIP2*, *SaZIP3*, *SaZIP4*, and *IRT1* genes, whereas in roots it upregulates *SaHMA3*, *SaNramp6*, *SaZIP2*, *SaZIP4*, *SaZIP11*, and *IRT1* genes.	[162]
Cd^2+^	*Buttiauxella* sp. strain SaSR13	*Sedum alfredii*	IAA production, phosphate solubilization, siderophore production, and ACC deaminase activity.	Bacterium enhances biomass accumulation, root growth, and root-surface area, increases photosynthetic pigments (Chl), and reduces superoxide anion levels.	[135]
Cd^2+^	*Pseudomonas veronii* strain E02	*Panicum virgatum*	IAA production and ACC deaminase activity.	Bacterium enhances biomass accumulation and increases stem growth.	[163]
Cd^2+^	*Pseudomonas rhodesiae* strains GRC065, GRC066, GRC093, GRC140	*Arabidopsis thaliana* Col-0	Phosphate solubilization, siderophore production, IAA, and ACC deaminase activity.	Bacteria promote the development of lateral roots in *A. thaliana* seedlings cultivated in conditions with and without cadmium.	[53]
Cd^2+^	*Enterobacter* sp. strain S2	*Oryza sativa*	ACC deaminase activity, IAA production, phosphate solubilization, and nitrogen fixation.	Bacterium enhances seedling growth, germination percentage, root-shoot length, fresh and dry weight, amylase, and protease activity. Furthermore, it exhibited alleviation of Cd-induced oxidative stress, reduction of stress ethylene, and decreased Cd accumulation in seedlings, conferring plant tolerance to cadmium.	[164]
Cd^2+^	*Pseudomonas fluorescens*	*Sedum alfredii*	IAA production, siderophore production, and ACC deaminase activity.	Bacterium promotes lateral root formation, enhances biomass, Cd uptake and accumulation, increases IAA concentration, and decreases abscisic acid, brassinolide, trans-zeatin, ethylene, and jasmonic acid in roots, thereby inducing lateral root emergence. Moreover, it activates plant hormone-related genes.	[105]
Cd^2+^	*Rhodococcus ruber* N7	*Sorghum bicolor*	ACC deaminase activity, siderophore, and IAA production.	Bacterium increases the activity of peroxidase, laccase, and tyrosinase. Under cadmium contamination, it successfully colonizes the roots and contributes to metal accumulation in the plant roots.	[165]
Cd^2+^	*Pseudomonas rhodesiae* strain GRC140	*Cucumis sativus* L.	Phosphate solubilization, siderophore production, ACC deaminase activity, IAA andphenylacetic acid (PAA) synthesis.	In Cd-exposed seedlings, the bacterium improves the growth of *C. sativus* L.	[166]
Cd^2+^	*Enterobacter cloacae* strain AS10	*Oryza sativa*	Phosphate solubilization, ACC deaminase activity, nitrogen fixation, siderophore, HCN, and IAA production.	Bacterium enhances root-shoot growth at the seedling stage through Cd immobilization. It increases total sugar content and prevents the surge of ethylene and oxidative stress.	[167]
Cd^2+^, Ni^2+^ and Pb^2+^	*Citrobacter werkmanii* strain WWN1	*Triticum aestivum* L.	Zn, K, and PO_4_ solubilization, siderophore production.	Bacteria enhance plant shoot and root length, fresh and dry weight, and photosynthetic pigments (Chl *a* and *b*) under HM stress. Moreover, it improves antioxidant activity.	[168,169]
	*Enterobacter cloacaecepa* strain JWM6				
Cr^6+^	*Pseudomonas* sp. strain NT27	*Medicago sativa*	Phosphate solubilization, siderophore production, IAA and HCN production.	Bacterium increases shoot and root dry weights in the presence of Cr. Increases chlorophyll content and decreases stress markers, malondialdehyde, hydrogen peroxide, and proline levels.	[170]
Cr^6+^	*Pseudomonas* sp. strain CPSB21	*Helianthus annuus* L. and *Solanum lycopersicum* L.	Phosphate solubilization, siderophores, IAA, HCN and ammonia production.	Bacterium enhances shoot and root length, fresh and dry weight, chlorophyll, and soluble protein content. It reduces adverse effects of metal stress.	[171]
Cr^3+^	*Bacillus cereus* strain B05	*Brassica nigra*	Phosphate solubilization, siderophore production, ACC deaminase synthesis, phytohormones (IAA, CK, ABA).	Bacterium promotes plant growth and reduces chromium toxicity. It enhances seed germination %, shoot and root length, fresh, and dry biomass, and photosynthetic pigments. It improves phytoextraction of Cr.	[172]

## 6. Other Microorganisms Associated with the Rhizosphere of *Typha* spp.

The rhizosphere is the habitat of many microorganisms and invertebrates, considered one of Earth’s most dynamic interfaces [104]. Although bacteria are the main inhabitants of the rhizosphere, the plant roots can be colonized by fungi, protozoa, rotifers, nematodes, and microarthropods, which are attracted by rhizodeposits, nutrients, exudates, border cells, and mucilage released by the plant root [100].

The presence of endophytic fungi has been reported in *T. latifolia* roots, including *Aspergillus* sp., *Myrothecium* sp., *Phoma* sp., *Penicillium* sp., *Acremonium* sp., and *Fusarium* sp., while in its rhizome *Penicillium* sp., *Myrothecium* sp., and *Fusarium* sp. have been found [173]. Furthermore, the efficiency of *Aspergillus niger*, *Acremonium* sp., *Aureobasidium* sp., *Cephalosporium* sp., and *Fusarium* sp., to degrade pollutants or absorb HMs has been demonstrated; thus, they are an option in mycoremediation [173,174]. On the other hand, a study conducted in Zhejiang Province, Eastern China, by Guan et al. [175] reported the presence of common fungal species including *Pleosporales* sp., *Teratosphaeria microspora*, *Geotrichum candidum, Engyodontium album, Blastocladiales* sp., uncultured soil fungus, *Fusarium graminearum*, and *Cladosporium bruhnei* in the rhizosphere of *T. latifolia* and *T. orientalis.* While *Paramicrosporidium saccamoebae, Rhynchosporium secalis*, *Acremonium roseolum*, and *Cladosporium sphaerospermun* were found in *T. latifolia*, and uncultured soil fungus in *T. orientalis*.

Finally, arbuscular vesicular (AV) fungi are known to increase plant resistance to heavy metals, and this differs according to fungal species, plant, and environmental conditions [176].

## 7. Conclusions

The *Typha* genus is used in phytoremediation because of its ability to remove heavy metals and accumulate them in its roots. This ability is enhanced by the action of bacterial communities associated with the plant roots. However, little is known about bacterial species that colonize the roots of *Typhaceas*.

Efforts have been made using microbiological and molecular strategies to identify either cultivable or non-cultivable bacteria associated with the root of *Typha* species grown in natural environments and wetlands contaminated with HMs. However, the information available is scarce, making it difficult to establish the core of bacterial communities associated with the roots of *Typhaceas*.

Data available in the GenBank database show that the rhizosphere of *T. latifolia* growing in natural wetlands is mainly colonized by bacteria belonging to *Proteobacteria*, *Actinobacteria*, and *Firmicutes*; among them are bacterial species classified as plant growth-promoting rhizobacteria that exert biochemical activities possibly involved in plants adapting to natural wetland conditions.

On the other hand, cultivable plant growth-promoting rhizobacteria have attracted much attention because they can be used in plant–bacteria–metal interaction assays. It has been demonstrated that endophytic bacteria isolated from *T. angustifolia* promote the growth of their host plant in the presence of Fe. Likewise, endophytic *Pseudomonas* species isolated from the roots of *T. latifolia* promote the development of their host plant in the presence of Cd or Pb. The effect of plant growth-promoting rhizobacteria during their plant interaction could be attributed to IAA production, phosphate solubilization, siderophore production, and ACC-deaminase activity, which modulate plant physiology to adapt to environmental conditions.

Although there are significant advances in the role of some root-endophytic bacteria in their *Typha* host, it is necessary to continue identifying bacteria associated with the roots of *Typha* species growing in heavy metal-contaminated environments in order to select the most tolerant bacterial isolates and combine them in consortia to evaluate their contribution to plant development and phytoremediation. Additionally, it is important to establish the *Typha*–bacteria interaction model to determine the mechanisms involved in promoting plant growth and phytoremediation improvement. If well-characterized consortia or bacterium are available, they could help us to assist *Typha* in removing heavy metals from contaminated sites.

## 8. Outlooks

Industrial or domestic effluents are known to contain a diversity and abundance of compounds, including HMs, organic matter, nutrients, and emergent contaminants. Their concentration can vary in function of annual stages, weather, drought, and rains. These variations influence plant physiology, growth and development, and the released root exudates. The chemical composition of root exudates influences the recruitment, composition, and abundance of bacteria inside the root endosphere or in the root rhizoplane of *Typha*. Therefore, the bacterial population’s structure can vary during the year in each microhabitat of the plant, and even along the wetland.

It has been suggested that plants select bacterial populations that improve their growth in the wetland’s conditions. Thus, it is essential to determine the population dynamics of non-cultivable and cultivable bacteria associated with the roots of *Typha* species growing in wetlands to treat wastewater. Although main plant-colonizer rhizobacteria have been identified, their behavior, ecological interactions, HM-tolerance mechanisms, biochemical abilities, PGPR activities, colonization site, abundance, and diversity remain poorly understood. Likewise, it is important to establish if bacterial populations are specific for the tolerance and removal of a specific heavy metal or whether a bacterial core community is involved in the tolerance and removal of multiple heavy metals. Finally, bacterial communities’ knowledge could help to select specific bacterial isolates for bioaugmentation, enhancing contaminant removal and the rational manipulation of plant–microbiota interactions.

## Figures and Tables

**Figure 1 microorganisms-11-01587-f001:**
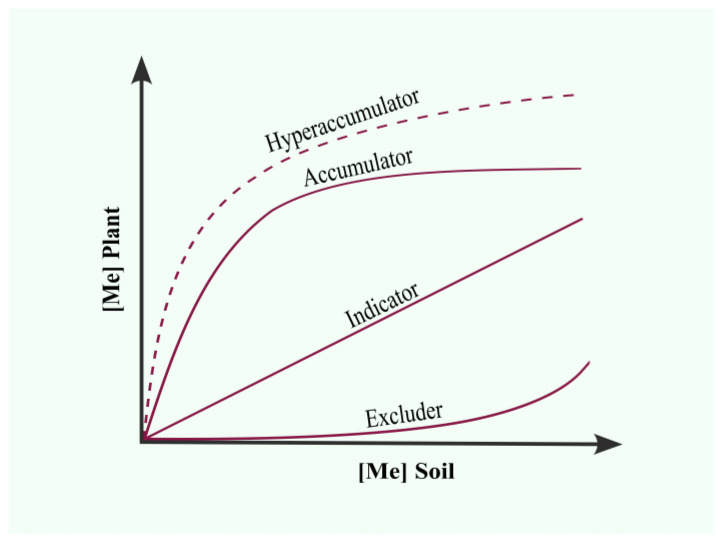
Classification of plants according to their heavy metal removal ability.

**Figure 2 microorganisms-11-01587-f002:**
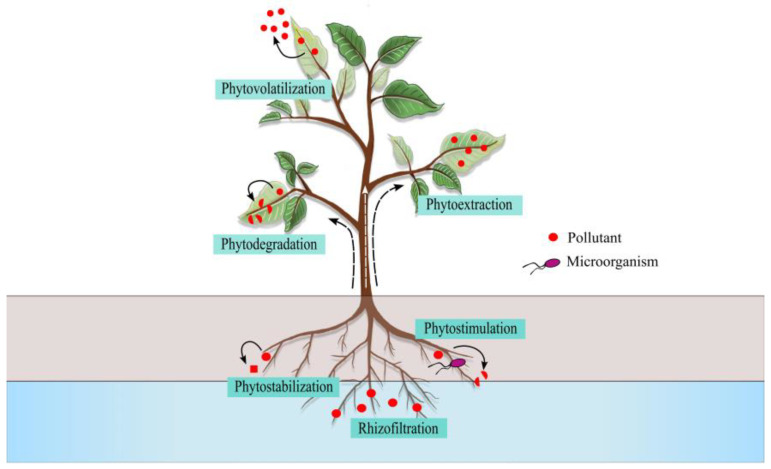
Phytoremediation mechanisms classification.

**Figure 3 microorganisms-11-01587-f003:**
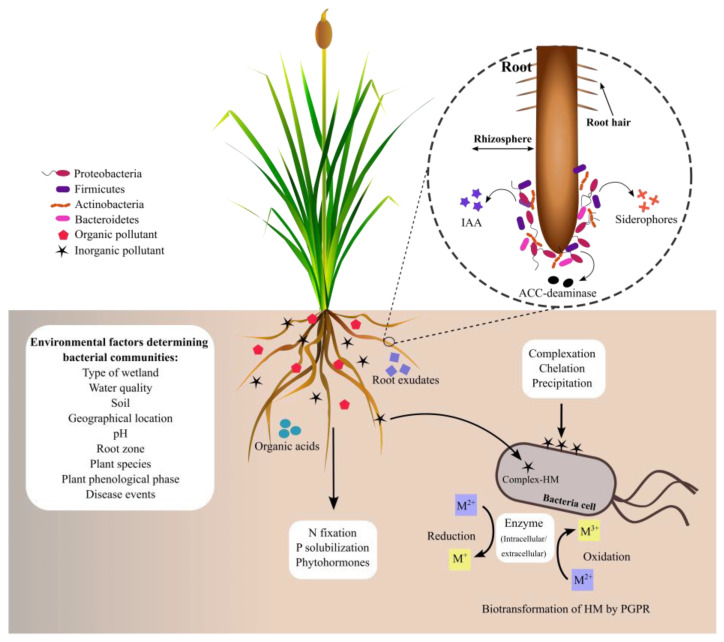
Bacterial phyla commonly associated with the plant rhizosphere. Bacterial phyla comprise species adapted to the rhizosphere conditions, where they interact among themselves by establishing complex interaction networks that contribute to the plant’s fitness to the environmental conditions. Biochemical activities of bacteria are discussed below.

**Figure 4 microorganisms-11-01587-f004:**
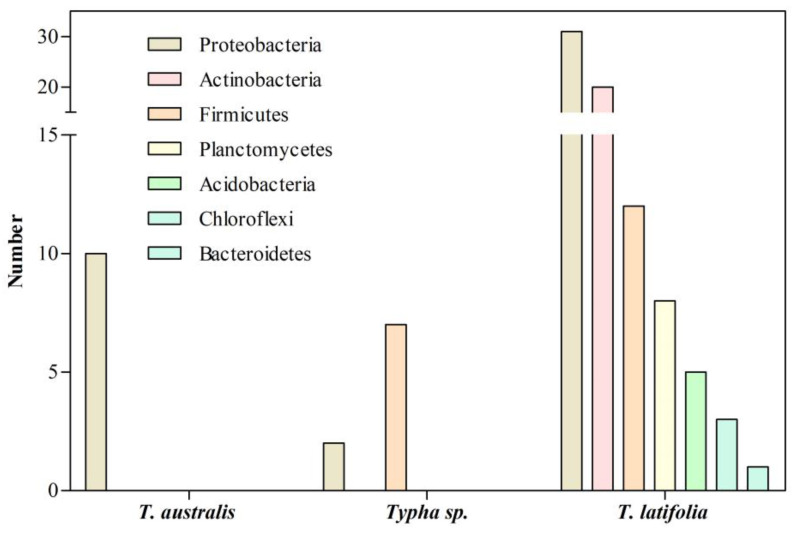
Bacteria associated with the roots of *Typha* species growing in natural environments.

**Figure 5 microorganisms-11-01587-f005:**
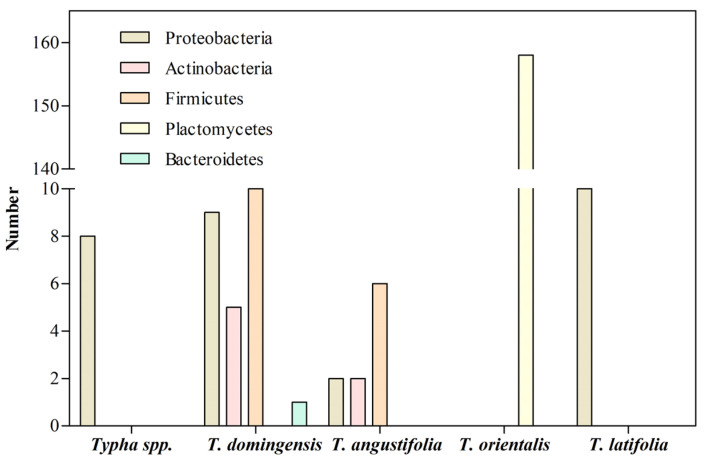
Bacteria associated with the roots of *Typha* species growing in presence of HMs.

**Table 1 microorganisms-11-01587-t001:** Mechanisms involved in phytoremediation processes.

Process	Mechanism	Contaminant
Phytoextraction	Hyperaccumulation	HMs, organic compounds, and radioisotopes
Phytovolatilization	Leaf volatilization	Organic compounds and Hg, As, and Se.
Phytostabilization	Precipitation, formation of insoluble complexes, valence reduction, and adsorption	HMs
Rhizofiltration	Accumulation in the rhizosphere	HMs and organic compounds
Phytodegradation	Enzymatic degradation	Organic pollutants
Phytostimulation	Microbial growth by stimulation	Organic pollutants

**Table 3 microorganisms-11-01587-t003:** Phyla associated with *Typha* exposed to heavy metals.

Specie	Phylum	Metal	Site	References
*Typha* sp.	Proteobacteria	Cr	Wetland	[69]
*T. domingensis*	Proteobacteria Firmicutes Actinobacteria Bacteroidetes	Cr, Ni, Fe	Pond and stream	[106]
*T. angustifolia*	Firmicutes Proteobacteria Actinobacteria	Fe	Wetland	[64]
*T. orientalis*	Planctomycetes Uncultured bacterium	Cu, Zn, Pb	Lake	[107]
*T. latifolia*	Proteobacteria	Cd	Contaminated site	[53]

## Data Availability

The data supporting this study’s findings are available from the corresponding author, A.H.-M., upon reasonable request.

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
