# Peer review of "Bacterial Communities Associated with the Roots of Typha spp. and Its Relationship in Phytoremediation Processes"

_microorganisms, 2023, doi:10.3390/microorganisms11061587_

Round 1

Reviewer 1 Report

This manuscript showed the role of Typha sp. in phytoremediation processes, the microbial diversity associated with the roots of plants exposed to heavy metal, and how bacterial communities contribute to the plant adaptation to contaminants and their participation in phytoremediation. This manuscript is very interesting. However, some aspects need to be further addressed.

Line 44, authors should tell us why choose the Typha genus for review? What is the special for these plants?

Line 108, “HM phytoremediation mechanisms”, this part seems not new for the phytoremediation.

Table 2, please provide the detail phytoremediation results for Typha species used in phytoremediation. Furthermore, N, and P were not belonged to HM.

Fig. 3 is too simple to present the commonly bacterial phyla associated with the plant rhizosphere.

Table 4, Cd+2 changed into Cd2+.

Author Response

Reviewer 1

This manuscript showed the role of Typha sp. in phytoremediation processes, the microbial diversity associated with the roots of plants exposed to heavy metal, and how bacterial communities contribute to the plant adaptation to contaminants and their participation in phytoremediation. This manuscript is very interesting. However, some aspects need to be further addressed.

 R= We appreciate the time you spent reviewing the document and the opportunity to answer the question to improve the review.

1.- Line 44, authors should tell us why choose the Typha genus for review? What is the special for these plants?

 Thanks for your suggestion. Please refer lines 44-52.

Typha spp. was chosen for its widespread distribution in aquatic and semiaquatic ecosystems around the world, and for its ability to persist in polluted environments. It has been used in constructed wetlands to treat industrial effluents containing heavy metals, drugs, nutrients, pesticides, and domestic wastewater, etc. Hence its importance as a study subject of this review.

2.- Line 108, “HM phytoremediation mechanisms”, this part seems not new for the phytoremediation.

Thanks for your suggestion. We update the references and include information to improve this section. Please refer lines 121-125, lines 130-133, lines 147-152, lines 166-180.

3.- Table 2, please provide the detail phytoremediation results for Typha species used in phytoremediation. Furthermore, N, and P were not belonged to HM.

Thanks for your suggestion. We improve the Table 2, including information about phytoremediation by Typha species. Please refer the information in Table 2.

4.- Fig. 3 is too simple to present the commonly bacterial phyla associated with the plant rhizosphere. 

Thanks for your suggestion. We improve the figure 3 and included detailed description of it. 

5.- Table 4, Cd+2 changed into Cd2+.

Thanks for your suggestion. Changes were made in Table 4.

Best regards

Alejandro Hernández Morales

Reviewer 2 Report

The interactions of plant-microbial is one of very important issues in both of life and environmental sciences. It is interesting that the authors have reviewed bacterial communities associated with the roots of Typha spp. and its relationship in phytoremediation processes. However, the MS need to be revised before published.

1.       Title, change to “bacterial communities associated with the roots of emergent wetland and its relationship in phytoremediation processes”. The topic needs to be expended, but Typha spp. can be as an example.

2.       It is lack of outlook for the future studies on the subjects. The review paper should point out clearly what study directions in the future.

3.       Change “arbuscular vesicular (VA)” to “arbuscular vesicular (AV)”.

Minor editing of English language required.

Author Response

Reviewer 2

The interactions of plant-microbial is one of very important issues in both of life and environmental sciences. It is interesting that the authors have reviewed bacterial communities associated with the roots of Typha spp. and its relationship in phytoremediation processes. However, the MS need to be revised before published.

 R= We appreciate the time you spent reviewing the document and the opportunity to answer the question to improve the review. 

1.- Title, change to “bacterial communities associated with the roots of emergent wetland and its relationship in phytoremediation processes”. The topic needs to be expended, but Typha spp. can be as an example. 

R= Thanks for your suggestion. The proposed title change is an excellent topic. Indeed, study of microbiome of macrophytes has been attracted much attention.

Piertangelo et al. (1) reported the composition of the root-associated bacterial microbiota of Phragmites australis and Typha latifolia. They demonstrated that the microbiota associated with the rhizosphere of P. australis and T. latifolia tends to converge toward a common taxonomic composition dominated by members of the phyla Actinobacteria, Firmicutes, Proteobacteria, and Planctomycetes.

Wang et al. (2) reported the Phragmites root-inhabiting microbiome: A critical review on its composition and environmental application. In this review summarizes the advances on the Phragmites root microbiome, including bacteria, archaea, and fungi.

Therefore, our working group decided to contribute with a review of the microbiota of the genus Typha because is an iconic wetland plant found in wetlands worldwide that is capable of rapidly colonizing habitats and forming monodominant vegetation stands due to traits such as robust size, rapid growth rate, and rhizomatic expansion. Also, it has high capacity to adapt to nutrient deprivation, flood, salinity, and drought.

1 Pietrangelo, L., Bucci, A., Maiuro, L., Bulgarelli, D., & Naclerio, G. (2018). Unraveling the composition of the root-associated bacterial microbiota of Phragmites australis and Typha latifolia. Frontiers in microbiology, 9, 1650.

2 Wang, D., Bai, Y., & Qu, J. (2022). The Phragmites root-inhabiting microbiome: A critical review on its composition and environmental application. Engineering9, 42-50.

2.- It is lack of outlook for the future studies on the subjects. The review paper should point out clearly what study directions in the future. 

R= Thanks for your suggestion. We include prospect section. Please refers lines 592-611. 

3.- Change “arbuscular vesicular (VA)” to “arbuscular vesicular (AV)”.

R= Thanks for your suggestion. Change was made.

Best regards

Alejandro Hernández Morales

Round 2

Reviewer 1 Report

I have no questions. I suggested that can be accepted by the journal.

 English Language is OK.

Reviewer 2 Report

This is a revised MS. The authors have corrected the reviewers' suggestions. Hence, it is recommended to be published in the present form.

 Minor editing of English language required.